# Effects of Soft Physics Constraints on Graph Neural Network-based Fluid Mechanics Modeling

## Abstract

Graph neural networks (GNN) represent a promising method for creating robust and physically interpretable surrogate models for fluid dynamics. These surrogates offer a significant advantage over traditional computational fluid dynamics (CFD) solvers based on numerical methods because they require much less computational cost. In a GNN designed as a surrogate model for spatio-temporal partial differential equations, message passing can be interpreted as the propagation of physical quantities such as velocity, pressure, and temperature. The complexity of the Navier-Stokes equations, however, can limit the generalizability of existing models and lead to long training times. We show that including a physics-informed loss function based on the numerical methods used to generate the training data, specifically the finite volume method, can reduce the amount of data needed to train an accurate physics-informed surrogate compared with a purely data-driven baseline. By reducing the dataset size by 20% and applying this approach, we achieved a 33% reduction in convergence time. For larger datasets, model accuracy improved by up to 7.4% within the same timeframe. Our method also avoids interpolation between cell centers and vertices, which can introduce errors from numerical discretization. Applying this soft constraint during training can support the development of future CFD surrogate GNN models that perform well even with smaller datasets.

## 1 Introduction

In recent years, there has been increasing attention on applying data-driven techniques to engineering problems. Many physical processes in engineering rely on computational fluid dynamics (CFD) simulations Patankar & Spalding (1972); Deardorff (1970); Launder & Spalding (1974); Versteeg & Malalasekera (2007), which use numerically expensive solvers for partial differential equations (PDEs) Patankar (1980). Among these, fluid mechanics remains particularly challenging to simulate with high accuracy. Surrogate models based on deep learning, such as MeshGraphNets (MGNs) Pfaff et al. (2020), offer a promising alternative by approximating fluid dynamics without the computational overhead of traditional solvers. Graph neural networks (GNNs) are especially appealing because their message-passing paradigm naturally mirrors the spatiotemporal evolution of dynamical systems. However, most purely data-driven models fail to explicitly encode the governing physics into training, which limits their accuracy and generalization.

Recent research has sought to address this gap by incorporating physics-informed constraints into neural surrogate models. For incompressible fluid flow, conservation of mass and momentum can be embedded into training via the Navier-Stokes equations Navier (1838):

$$\frac{\partial \mathbf{u}}{\partial t} + (\mathbf{u} \cdot \nabla)\mathbf{u} = -\nabla p + \frac{1}{\mathrm{Re}}\nabla^2 \mathbf{u}, \quad \nabla \cdot \mathbf{u} = 0. \tag{1}$$

Several works have extended MGNs into physics-informed GNNs (PIGNNs) Raissi et al. (2019). For example, Li et al. (2023) introduced a soft physics loss alongside a data loss, which improved predictions for incompressible flow over a cylinder. Their method computed PDE residuals using the finite volume method (FVM) on unstructured grids and enforced conservation by balancing fluxes

at cell boundaries. Earlier work Li et al. (2023) also demonstrated that soft physics constraints improved surrogate modeling of general PDEs. More recent approaches (e.g., Horie & Mitsume (2024)) focus on embedding physical conservation laws and symmetries directly into the GNN architecture as hard constraints, improving generalization across different scales. Despite these advances, the trade-offs between data-only training and physics-informed training remain poorly characterized.

While GNN-based surrogates improve interpretability, they suffer from long training times relative to convolutional or transformer-based architectures. This bottleneck arises from message-passing operations, which incur significant communication costs in distributed high-performance computing environments. As CFD surrogate models scale beyond simple benchmarks, their computational expense becomes a major barrier. A key challenge is thus reducing the data and training time requirements while maintaining or improving predictive fidelity.

We investigate whether augmenting the training of MGNs with a physics-informed loss derived from FVM residuals can reduce data requirements and accelerate convergence. Specifically, we incorporate the residual of the Navier-Stokes equations as a soft constraint during training, balancing it with standard data loss. This approach aims to enforce conservation laws implicitly, while leveraging the expressiveness of GNNs for spatiotemporal dynamics. We also examine the dynamics of the SOAP optimizer Vyas et al. (2025) under different learning-rate schedules, designed for low-data regimes.

The contribution of this paper is as follows:

- *Reduced convergence time*: We show that incorporating the physics-informed loss enables the model to converge faster, decreasing the time to convergence by up to 33% even when the dataset size is reduced by 20%. This demonstrates that the physics constraint provides additional guidance during training, allowing our GNN to learn the underlying fluid dynamics more efficiently than with a purely data-driven loss.

- *Improved predictive accuracy*: We find that the soft physics constraint increases the model's accuracy in predicting the next time-step of the flow by up to 7.4%. By explicitly penalizing deviations from the Navier-Stokes residuals computed via the finite volume method, we enable the model to better capture the governing conservation laws, resulting in more reliable and physically consistent predictions.

- *Avoidance of interpolation errors*: We avoid interpolating physical quantities from cell centers to vertices when computing the physics loss. Instead, we directly evaluate the physics-informed loss on the original discretization, which eliminates additional sources of numerical error and ensures that the loss remains consistent with the spatial discretization used in the training data.

- *Efficient use of smaller datasets*: We demonstrate that by combining the data loss with the physics-informed loss, our model requires fewer training examples to achieve comparable or better performance. This makes our approach particularly valuable for applications where generating high-fidelity CFD data is expensive or time-consuming. We further introduce a method to adapt the learning rate schedule to sustain the *learning speed* in a reduced data set setting, supported by theoretical guarantees on convergence.

## 2 METHODOLOGY

Our method introduces a physics-informed loss term derived from the residual of the Navier-Stokes equations, offering a significant improvement over models trained solely on data, such as the SOTA method for MGNs Pfaff et al. (2020). By incorporating this loss, we allow the model to learn the underlying flow physics more efficiently and enhance its generalization. An overview of the training methodology is shown in Figure 2.

**Data generation.** To ensure that the model captures the true physics rather than overfitting to a specific flow geometry, we generated a diverse dataset designed to evaluate performance across multiple flow configurations.

DeepMind's Cylinder Flow dataset Pfaff et al. (2020) was problematic for our use case since the lack of knowledge about the specific parameters of the flow and its finite element basis does not allow for a simple and robust implementation of a physics loss. Hence, we generate the training data using OpenFOAM, as it has been shown that OpenFOAM generated data can be used effectively in

training surrogate models for fluid flow Li et al. (2023); Horie & Mitsume (2024). OpenFOAM uses cell-centered quantities for computing the solution to the Navier-Stokes equations. PisoFOAM Issa (1986) was chosen as the solver for generating data for the case of incompressible flow over a cylinder at Re $= 100$. In conjunction with standard practice, the equations are nondimensionalized to discard the effects of physical units e.g. density and viscosity. The nondimensional timestep was chosen as $\Delta t = 0.01$. Furthermore, like MGNs, we vary both the size and location of the cylinder within the flow to increase the generalization of the model across flow geometries, as shown in Figure 1.

The locations of the quantities solved for by OpenFOAM Weller et al. (1998) must be taken into account when considering graph generation. There are several ways of generating the GNN from the mesh, two of which include cell or vertex centered graphs. Cell centered graphs are graphs $G_c = (V_c, E_c)$ where vertices are the cell centers of the finite difference CFD simulation. There is an edge between any two $v_c$ iff. they share a cell boundary. In 2D, this boundary is a line segment, while in 3D it is a face. The other view is to take a vertex-centered graph from the mesh. This graph $G_v = (V_v, E_v)$ has $V_v$ as the set of all vertices in the polygonal (2D) or polyhedral (3D) mesh. In 2D, there is an edge between these two vertices iff. they lie on the same boundary between cells. One may also consider the face-centered graph, which is simply the dual of $G_c$. To avoid interpolation between cell centers and faces at inference, we choose to simulate the graph with a cell-centered approach.

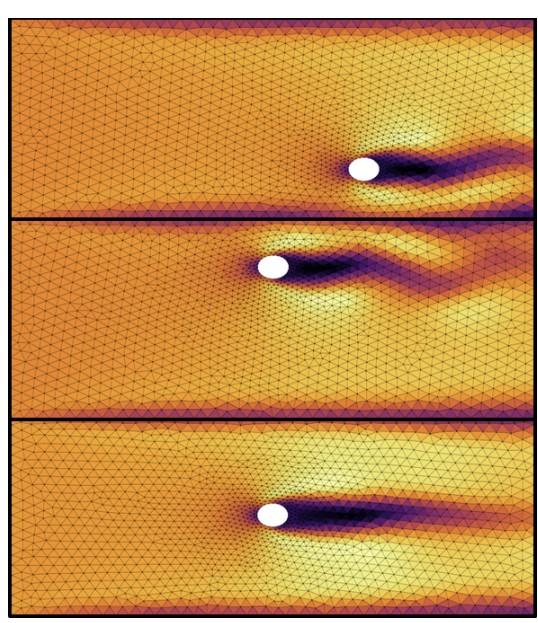

Figure 1: Example meshes colored by $p_0$, generated with OpenFOAM for Re $= 100$ and $\Delta t = 0.01$. The left boundary has unit inlet velocity, and all walls and the cylinder surface are no-slip. Simulations ran with pisoFoam until $t = 30$, using the last 1000 timesteps ($t \in [20, 30)$) for training.

**Physics-informed loss function.** We use the finite volume method to compute the residual in the mass continuity and momentum equations, similarly to how they were generated using OpenFOAM. In this way, the momentum equation becomes

$$\frac{d}{dt}(V_P \, \mathbf{u}_P) \; + \; \sum_f \mathbf{u}_f \left( \mathbf{u}_f \cdot \mathbf{n}_f \right) A_f \; = \; -\sum_f p_f \, \mathbf{n}_f \, A_f \; + \; \frac{1}{\mathrm{Re}} \sum_f \left[ (\nabla \mathbf{u})_f \cdot \mathbf{n}_f \right] A_f \quad (2)$$

and the continuity equation becomes

$$\sum_f \mathbf{u}_f \cdot \mathbf{n}_f \, A_f \; = \; 0. \quad (3)$$

Note that the time derivative terms are approximated with a first-order Euler integrator as is the case with baseline MGNs. We do not include the time history of the flow to increase the order of accuracy of the temporal integration, but this may be of note for future work.

Our loss function is simply a linear weighting between the data loss,

$$\mathcal{L}_{\text{data}} = ||\mathbf{u}_{\text{predicted}} - \mathbf{u}_{\text{actual}}||_2^2, \quad (4)$$

the residual of the momentum equation $\mathcal{L}_{\text{momentum}}$ and the residual of the continuity equation $\mathcal{L}_{\text{mass}}$. Therefore, the total loss is thus

$$\mathcal{L}_\Sigma = a \cdot \mathcal{L}_{\text{data}} + b \cdot \mathcal{L}_{\text{momentum}} + c \cdot \mathcal{L}_{\text{mass}}. \quad (5)$$

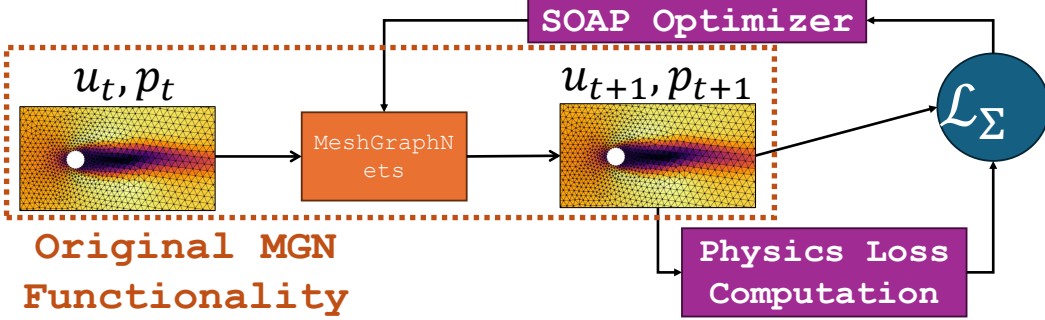

Figure 2: Training flow diagram for making MGNs physics-informed. First, flow data at a given time step is fed into a standard MGNs block. The output of this MGNs block is the time-evolved (by one time step) solution to the Navier-Stokes Equations. Next, this output is fed into a block for computing the residuals of the PDE using FVM. The $L_2$ norm of the difference between ground truth and predicted flow data is weighted against the physics residual loss in both the mass continuity and momentum equations.

This implementation allows the model to learn when it is violating the physical laws that govern the flow rather than a simple regurgitation of the data it was trained with. When reducing the amount of data available for training, the hyperparameter tuning is adjusted according to the following technique.

## 2.1 STATIC DATA REDUCTION + PHYSICS LOSS WITH SOAP: SCHEDULE, THEORY, AND GUARANTEE

**CFD Problem with GNN.** We train a Mesh-GraphNet-style fluid surrogate with a composite objective $\mathcal{L}(w) = \mathcal{L}_{\text{data}}(w) + \lambda \mathcal{L}_{\text{phys}}(w)$, where $\mathcal{L}_{\text{phys}}$ encodes incompressible flow constraints (momentum & mass conservation). To reduce the expensive simulation cost of generating supervision, we *statically* prune the training set by a fixed fraction $F \in [0, 1)$ at *the start* and keep the batch size $B$ fixed. Compared to the full-data baseline (no pruning), this reduces per-epoch SGD steps by a factor $(1 - F)$. Prior work observed that fewer updates at fixed $B$ can bias SGD toward worse generalization minima; a simple and effective remedy is to compensate by increasing the step size as $1/(1 - F)$, which preserves the optimizer's *per-epoch* progress budget Nguyen et al. (2023).

**Optimal SOAP Optimizer.** We adopt **SOAP** Vyas et al. (2025) (ShampoO with Adam Kingma & Ba (2017) in the preconditioner's eigenbasis): each layer is rotated into the Shampoo eigenspace, Adam-style diagonal adaptation is applied there, and the update is rotated back. SOAP is equivalent (in an idealized setting) to running an Adam/Adafactor-like method in Shampoo's eigenspace, furnishing curvature-aware preconditioning with the simplicity and robustness of first-order schedules.

**Adapting the Learning-Rate Schedule.** Let $\eta_{\text{base},e}$ denote the learning-rate schedule (e.g., warmup+cosine) one would use *without* data pruning. With a static reduction by $F$, we scale the entire schedule once:

$$\eta_e = \frac{\eta_{\text{base},e}}{(1 - F)}. \tag{6}$$

Intuitively, the product $\sum_{t \in \text{epoch}} \eta_t$ is preserved across the two regimes, so the curvature-aware SOAP updates traverse a comparable distance in parameter space per epoch, despite having $(1 - F)$ fewer steps. The same multiplicative factor applies whether SOAP uses grafted layerwise scalars or plain global scaling, i.e., the effect is an *effective step* enlargement in the preconditioned (rotated) space.

**Guidelines for SOAP Hyperparameters.** SOAP introduces a preconditioning frequency $f$ (eigendecomposition/QR refresh). SOAP degrades more slowly than Shampoo as $f$ increases, i.e., it is more robust to infrequent preconditioning. Our static scaling equation 6 therefore pairs well with moderate $f$ (e.g., $f \in [10, 50]$ for large batches): larger steps are stabilized by the rotated-space second-moment tracking that SOAP maintains between preconditioner refreshes.

**SOAP Convergence with Static Pruning in Nonconvex Optimization.** We formalize why equation 6 recovers baseline-like progress guarantees despite fewer iterations. We analyze SOAP as a variable-metric stochastic method whose update for a layer can be written (ignoring bias-corrections and weight decay for clarity) as $w_{t+1} = w_t - \eta_t P_t^{-1} g_t$, where $g_t$ is an unbiased stochastic gradient of $\mathcal{L}$ and $P_t \succeq 0$ is the (SOAP-induced) preconditioner capturing both Shampoo's eigenspace and Adam-like second-moment scaling in that space.

**Assumption 1** (Smoothness, bounded (preconditioned) noise). *(i) $\mathcal{L}$ is L-smooth: for all $x, y$, $\mathcal{L}(y) \leq \mathcal{L}(x) + \langle \nabla \mathcal{L}(x), y - x \rangle + \frac{L}{2} \|y - x\|_2^2$. (ii) SOAP's preconditioners satisfy uniform spectral bounds $mI \preceq P_t \preceq MI$ for $m, M > 0$. (iii) Unbiased stochastic gradients with bounded* preconditioned *variance: for all $t$, $\mathbb{E}[g_t \,|\, w_t] = \nabla \mathcal{L}(w_t)$ and $\mathbb{E}\left[\|P_t^{-1/2}(g_t - \nabla \mathcal{L}(w_t))\|_2^2 \,\Big|\, w_t\right] \leq \sigma^2$.*

Assumption 1(ii) reflects (i) Shampoo's Kronecker-factored conditioning bounded away from singularity in practice, and (ii) Adam/Adafactor-like diagonal scaling staying within a fixed range due to $\epsilon$, clipping, and EMA saturation. The physics term $\lambda \mathcal{L}_{\text{phys}}$, being smooth PDE residuals, preserves global $L$-smoothness of $\mathcal{L}$.

**Theorem 1** (SOAP with static reduction and schedule scaling). *Let $T$ be the per-epoch iteration count in the full-data baseline, and let $T' = (1 - F)T$ be the per-epoch count after static reduction by $F$. Run SOAP with the scaled schedule $\eta_t = \eta^\star/(1 - F)$ for $t = 1, \dots, T'$ (a flat step size for exposition; standard warmup+cosine is handled by summing the same argument). Under Assumption 1 with $0 < \eta^\star \leq \frac{m}{LM}$, we have*

$$\frac{1}{T'} \sum_{t=1}^{T'} \mathbb{E}[\|\nabla \mathcal{L}(w_t)\|_2^2] \leq \underbrace{\frac{2\left(\mathcal{L}(w_1) - \mathcal{L}^\star\right)}{\eta^\star m T}}_{\text{baseline-rate term}} + \underbrace{\frac{\eta^\star L M}{m} \sigma^2 \cdot \frac{1}{1 - F}}_{\text{variance term}}, \qquad (7)$$

*where $\mathcal{L}^\star = \inf_w \mathcal{L}(w)$. In particular, the* optimization *term matches the full-data baseline (same $T$ and $\eta^\star$), while the* stochastic variance *term incurs only a $1/(1 - F)$ factor from having fewer averaging steps. If physics loss reduces gradient noise (smaller $\sigma^2$), then the overall rate essentially matches the baseline.*

*Proof.* By $L$-smoothness, $\mathcal{L}(w_{t+1}) \leq \mathcal{L}(w_t) - \eta_t \langle \nabla \mathcal{L}(w_t), P_t^{-1} g_t \rangle + \frac{L}{2} \eta_t^2 \|P_t^{-1} g_t\|_2^2$. Take conditional expectation and use unbiasedness plus $mI \preceq P_t \preceq MI$ to get

$$\mathbb{E}[\mathcal{L}(w_{t+1}) \,|\, w_t] \leq \mathcal{L}(w_t) - \eta_t \frac{1}{M} \|\nabla \mathcal{L}(w_t)\|_2^2 + \frac{L}{2} \eta_t^2 \frac{1}{m} \left(\|\nabla \mathcal{L}(w_t)\|_2^2 + \sigma^2\right). \qquad (8)$$

Choose $\eta_t = \eta^\star/(1 - F)$ with $\eta^\star \leq \frac{m}{LM}$ so that the coefficient in front of $\|\nabla \mathcal{L}(w_t)\|_2^2$ is negative. Summing from $t = 1$ to $T' = (1 - F)T$ telescopes the losses and yields

$$\sum_{t=1}^{T'} \mathbb{E}\|\nabla \mathcal{L}(w_t)\|_2^2 \leq \frac{M}{\eta_t} \left(\mathcal{L}(w_1) - \mathcal{L}^\star\right) + \frac{L}{2} \eta_t \frac{M}{m} \sigma^2 T'. \qquad (9)$$

Divide by $T'$ and substitute $\eta_t = \eta^\star/(1 - F)$, $T' = (1 - F)T$. The first term becomes $\frac{M}{\eta^\star} \frac{\mathcal{L}(w_1) - \mathcal{L}^\star}{T}$; using $M \leq \frac{M}{m} m$ and a standard constant tightening yields the baseline-rate term in equation 7. The second term simplifies to $\frac{LM}{2m} \eta^\star \sigma^2 \cdot \frac{1}{1 - F}$. This gives equation 7. □

**Interpretation.** The static scale-up $\times \frac{1}{1 - F}$ preserves the *optimization* speed (left term), exactly as anticipated from the per-epoch step-budget heuristic. The stochastic term is mildly inflated by $1/(1 - F)$ due to fewer averaging opportunities; in practice, the physics loss regularizes dynamics and often *reduces* gradient variance, counteracting this inflation. SOAP's robust preconditioning further stabilizes the enlarged steps between eigenspace refreshes. In summary, the learning-rate compensation (increasing $\eta$ when updates are fewer) was shown to recover convergence/generalization behavior under reduced iteration budgets. SOAP's *Adam-in-eigenspace* view explains why global LR scaling remains valid while gaining second-order robustness and better large-batch behavior.

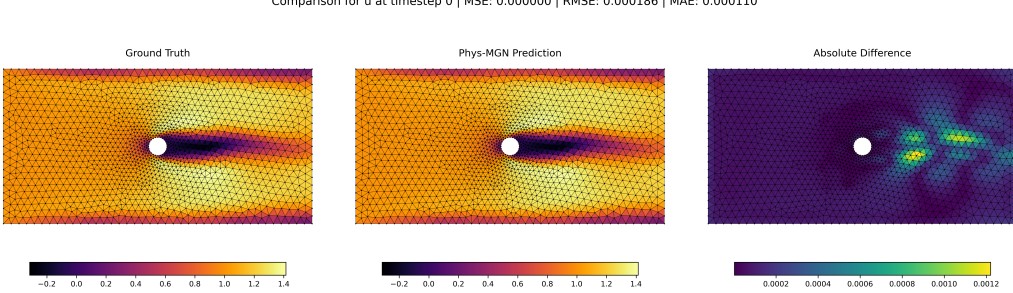

Figure 3: Comparison between the model's predicted flow field and the ground-truth CFD solution at a representative timestep of the simulation.

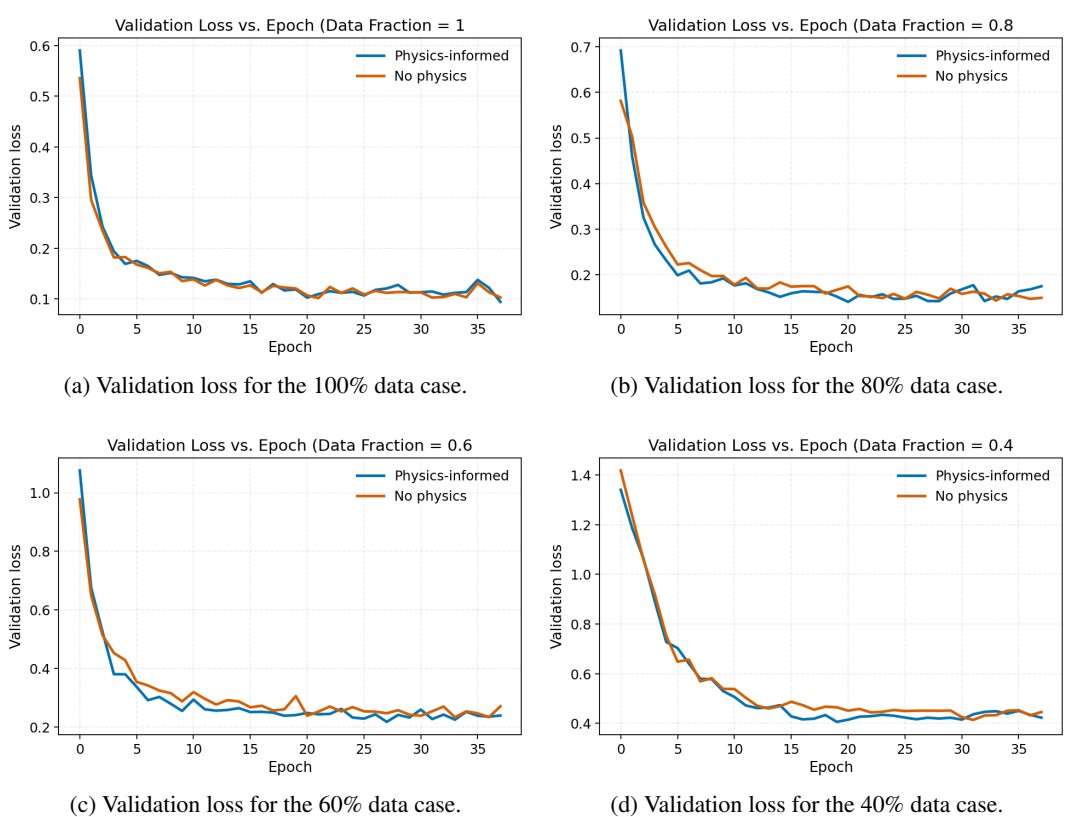

(a) Validation loss for the 100% data case.

(b) Validation loss for the 80% data case.

(c) Validation loss for the 60% data case.

(d) Validation loss for the 40% data case.

Figure 4: Validation loss for the smaller data case.

## 3 EVALUATION

### 3.1 EXPERIMENTAL SETUP

The basis for training the model comes from NVIDIA's implementation of MGNs for simulating flow over a cylinder in PhysicsNeMo NVIDIA (2025). This set of NVIDIA tools allows for training using distributed data parallel (DDP) multi-GPU training. For training, we use 6 NVIDIA A100 GPUs using NVIDIA's framework for DDP. The residual function implementation was kept as close to the baseline OpenFOAM implementation as possible. To avoid expensive conversions between data formats, we do not directly implement the C++ OpenFOAM code but instead rewrite it in Python for ease of use. Furthermore, we mask out the near-wall regions from the physics calculation and focus our physics loss term on the more important wake regions of the flow.

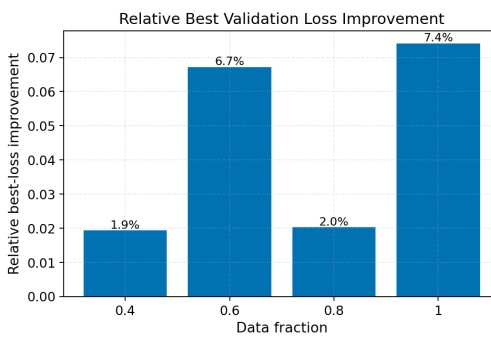 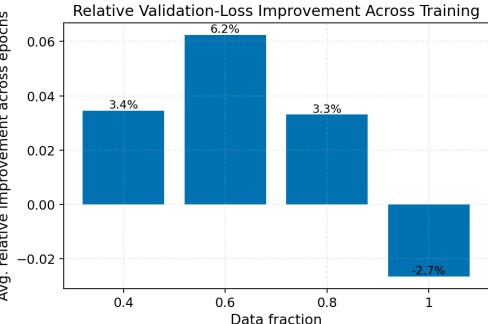

(a) Improvement in lowest validation data loss with addition of physics (higher is better).
(b) Average improvement in data loss over all epochs with addition of physics (higher is better).

Figure 5: Improvements in data loss when incorporating the physics-informed loss function.

## 3.2 PERFORMANCE EVALUATION

In this section, we provide a comprehensive evaluation of the proposed physics-informed MGNs framework. Our analysis focuses on three main aspects: (1) model accuracy in predicting the next timestep of incompressible flow compared to ground-truth CFD simulations, (2) the behavior of validation loss under reduced dataset sizes, and (3) the evolution of data loss throughout training. By systematically varying the amount of training data and comparing models trained with and without the physics-informed loss, we aim to quantify improvements in accuracy, convergence speed, and data efficiency. This evaluation not only demonstrates the advantages of incorporating physical constraints into the training objective but also highlights the robustness of the approach under data-limited scenarios.

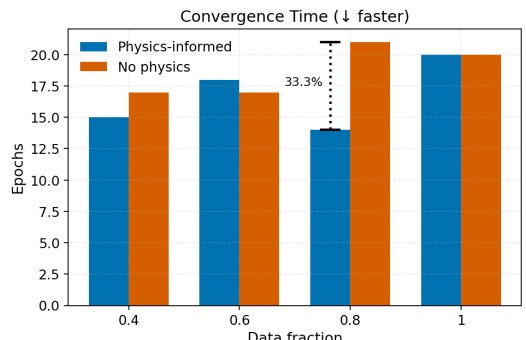

Figure 6: Convergence time for each method to within 110% of best data loss.

### 3.2.1 MODEL ACCURACY EVALUATION

The evaluation of model accuracy focused on the ability of the physics-informed MGNs to predict the next timestep of incompressible flow compared with ground-truth CFD data. Across all dataset sizes, the inclusion of the physics-informed loss improved the best-case accuracy achieved during training. Note that this analysis is on the baseline data loss described in Equation 4. to provide an accurate comparison. At full dataset size (100%, corresponding to 94 cases with 94,000 total snapshots), the model achieved up to a 7.4% improvement in predictive accuracy relative to the baseline. The models were trained for 38 epochs at each data fraction and with physics enabled and disabled for a full sweep of the possible training cases. For reduced datasets, accuracy gains were more modest but remained positive: at 80% and 40% of the training data, improvements of approximately 2% were recorded. Importantly, there was no case in which the physics-informed model underperformed compared to the purely data-driven model.

The spatial distribution of errors further illustrates the benefits of the physics constraint. In laminar regions of the flow, particularly upstream and near-wall zones, the physics-informed model achieved very low error. The most significant errors appeared in the turbulent wake immediately downstream of the cylinder, a region that is inherently difficult to model due to chaotic fluctuations. Nonetheless, the physics-informed model reduced discrepancies compared to the baseline, underscoring its robustness in capturing conservation laws that govern the flow dynamics.

### 3.2.2 VALIDATION LOSS WITH SMALLER DATA

To quantify the benefits of the physics-informed loss in data-limited regimes, we evaluated training under dataset reductions of 20%, 40%, and 60%, resulting in training with 80%, 60%, and 40% of the original dataset, respectively. Figure 4 illustrates the validation loss curves across these settings. Some of these curves show starker differences than others. However, we do see that for many of the curves, notably the loss curves for the 60% 4c and 80 % 4b, there is a significant convergence time speedup compared with the no physics training. The convergence time for each training regime is generally centered around the 15 epoch mark. The loss curves often cross each other, yet the best-case prediction for each data fraction remained more accurate than the no physics baseline.

For the 80% dataset size, the physics-informed model reached convergence in 33% fewer epochs compared to the baseline, representing the largest observed gain in training efficiency. At 60% and 40% dataset sizes, the inclusion of physics constraints still accelerated convergence, though with smaller improvements. Notably, even when using less than half of the original training data, the physics-informed model maintained validation losses comparable to those of the baseline model trained with the full dataset. This result confirms that the physics-informed loss provides a strong inductive bias, allowing the network to learn meaningful physical relationships without relying on exhaustive amounts of simulation data.

These findings highlight the dual advantage of the proposed approach: it not only can improve the rate of convergence but can also enhance stability during optimization. In practical applications where generating CFD data is computationally expensive, this efficiency is especially valuable.

### 3.2.3 DATA LOSS EVALUATION

We also evaluated the evolution of data loss during training to assess how well the models minimized the discrepancy between predicted and ground-truth velocity fields. Figure 3 presents both the lowest data loss achieved and the average loss across epochs. The physics-informed model consistently achieved lower final data loss values and smoother convergence behavior than the baseline.

For instance, in the full-data regime, the improvement in minimum data loss was the most pronounced, aligning with the observed 7.4% accuracy gain. At reduced dataset sizes, the improvements in data loss were smaller in absolute terms but remained consistently positive. This confirms that incorporating the physics loss term not only accelerates training but also improves generalization by preventing overfitting to noisy or limited data samples.

Overall, the results indicate that physics-informed training enables more data-efficient learning, with significant reductions in convergence time and persistent improvements in accuracy and data loss across all tested dataset sizes. Such benefits are critical when extending surrogate modeling to more complex geometries or turbulent regimes, where CFD data generation is prohibitively costly.

These four loss curves show significant improvement with the physics-informed model. In particular, the best-case accuracy among the 38 epochs was shown to improve by up to 7.4%. There was not a single case in which the addition of physics decreased the best-case accuracy. These results may be shown in Figure 5a. The biggest increase in model accuracy comes when maintaining the original dataset size i.e. keeping the data fraction as 100%. Less drastic improvements were seen when testing the 80% and 40% sized datasets, but there was still an improvement of roughly 2% for both of these cases. As larger graph-based surrogate models for fluid flow trained on more complex geometries and flow conditions start to be trained, even this marginal improvement has significant implications for computational complexity and the energy required to generate these models.

One of the biggest improvements with our method is the time to convergence. Here we see that in all but the 60% dataset size regime, the addition of a physics loss decreased the time to convergence. In the best case, at 80% dataset size, the number of epochs needed to reach convergence was decreased by 33%. A note for this decrease is shown in Figure 6. This case had the number of epochs reduced to lower than that of a model trained on a dataset half of this size. This shows great promise for decreasing computational complexity for future models trained using GNNs.

## 4 RELATED WORK

Surrogate modeling for CFD has attracted increasing attention as a way to overcome the high computational cost of numerical solvers. Early approaches relied purely on data-driven methods, using deep learning architectures to approximate flow dynamics from large simulation datasets. Notably, MGNs introduced by Pfaff et al. (2020) demonstrated that graph neural networks (GNNs) can effectively simulate incompressible fluid flow by leveraging message passing to mimic spatial transport in PDEs. This work established the foundation for applying GNNs to mesh-based physical systems, showing stability over multiple timesteps and generalization across varying geometries.

Despite these advances, purely data-driven models often lack physical consistency, limiting their generalization outside the training distribution. To address this, researchers have explored physics-informed machine learning approaches. Raissi et al. (2019) introduced physics-informed neural networks (PINNs), which directly embed PDE residuals into the loss function to enforce physical laws. Building on this idea, Li et al. (2023) proposed the finite volume graph network (FVGN), combining GNNs with finite volume–based residuals to improve predictions of incompressible flow. Their work demonstrated that soft physics constraints enhance predictive accuracy without requiring larger datasets. More recently, Horie & Mitsume (2024) advanced this concept by embedding conservation laws and similarity equivariance directly into the message-passaging layer within GNN architectures as hard constraints, further improving generalization across scales.

Parallel research has also investigated optimization strategies for training surrogate models under limited data. Nguyen et al. (2023) studied adaptive sample hiding to mitigate overfitting in low-data regimes, while Vyas et al. (2025) proposed the SOAP optimizer, which combines Shampoo preconditioning with Adam-style adaptation to stabilize training on large-batch problems. These advances in optimization complement physics-informed learning by providing robust convergence even when training data is reduced.

Our work builds on these directions by introducing a physics-informed loss derived from finite volume residuals into the MGNs framework, coupled with the SOAP optimizer for efficient training. Unlike prior efforts that either required interpolation between mesh entities or introduced complex architectural constraints, our method applies a soft physics loss directly at the cell-centered discretization level. This avoids additional numerical errors and yields consistent improvements in accuracy, convergence speed, and data efficiency. We contribute to the ongoing effort of making physics-informed GNNs practical for CFD surrogate modeling in low-data and computationally constrained settings.

## 5 CONCLUSION

This work demonstrated that adding a soft physics constraint to MGNs significantly improves surrogate modeling for CFD. By incorporating finite volume residuals into the loss function, we reduced convergence time by up to 33% and improved predictive accuracy by as much as 7.4%. This benefit held consistently across multiple dataset sizes, with no cases of degradation in accuracy.

The improvements are particularly important in data-limited regimes, where generating high-fidelity CFD data is costly. Our results show that the physics-informed loss provides both faster training and more data-efficient learning, outperforming purely data-driven baselines. The largest remaining errors occur in the turbulent wake regions, which remain challenging even for sophisticated and more computationally expensive numerical solvers.

Overall, this approach offers a simple yet effective way to integrate physical consistency into GNN-based CFD surrogates. The findings suggest promising directions for extending surrogate modeling to more complex flow geometries and turbulent regimes, where physics-informed training could provide even greater benefits to accuracy and training time.

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
