# OpenReview forum: "From Numerical Solvers to Graph Surrogates: Physics-Informed Losses for Data-Efficient CFD Modeling"
_ICLR.cc/2026/Conference — Submitted to ICLR 2026_

### Official Review · Reviewer_4MZT · 2025-10-27

**Soundness:** 2
**Presentation:** 2
**Contribution:** 2
**Rating:** 2
**Confidence:** 3

**Summary:**

Authors enhance MeshGraphNets by physics-informed loss function based on finite-volume method. The approach significantly reduces convergence time by up to 33% and improves predictive accuracy by up to 7.4% on dataset generated by authors using OpenFOAM.  It also allows for training with smaller datasets. The authors test these benefits across various data scales and demonstrate that the method is able to create more efficient and physically consistent GNN-based surrogate models.

**Strengths:**

1. Authors integrated FVM residuals into the GNN loss function and achieved improved predictive accuracy and more physically consistent simulations.
2. Convergence time was reduced by up to 33%, it is beneficial in data-limited scenarios.
3. Less training data is needed.
4. Avoids numerical errors from data interpolation by using cell-centered discretization.

**Weaknesses:**

Mostly, the paper feels largely incomplete.

1. Firstly, the title of paper in pdf is different from openreview: "Effects of soft physics constraints on graph neural network-based fluid mechanics modeling" vs "From Numerical Solvers to Graph Surrogates: Physics-Informed Losses for Data-Efficient CFD Modeling".  Although the meaning is probably the same, it is a bit confusing.
2. Generalization ability is limited. The entire study focuses on incompressible flow over a cylinder with fixed Reynolds number. The paper acknowledges that "The largest remaining errors occur in the turbulent wake regions", which are inherently difficult to model. This raises questions about how well the method would generalize to more complex scenarios.
3. "No physics" baseline definition. While "no physics" is clear in principle, the authors need to explain whether the baseline models were optimized with the same SOAP optimizer and learning rate scheduler. If it is not true, some of the performance gains attributed to physics could be partially influenced by optimization techniques.
4. The paper states that the loss function is "simply a linear weighting" and defines the coefficients a, b, c (equation 5). However, there's no discussion on how these weights were determined (and their values are not mentioned), if they were tuned, or their sensitivity to different flow conditions / dataset sizes. The choice of these weights can significantly impact the balance between data fidelity and physical consistency.
5. We don't know exactly how dataset pruning was done. Do authors drop samples randomly? Or do they drop some simulations based on some criteria? This is not stated on paper.
6. Strong justification of choosing SOAP optimizer is needed.
7. Provide explanation about why there are not direct comparisons with some previous studies mentioned in "Related work" section.

**Questions:**

1. Please fix the title in pdf or in openreview.
2. Provide more scenarios where you train and evaluate your approach. It is needed to better show generalization ability.
3. See W3. Explain what does "no physics" mean. Is it optimized with the same SOAP optimizer or it is vanilla MeshGraphNets.
4. Provide exact values of coefficients a,b,c in equation 5 with the procedure how they were determined. Ideally, do ablation study with different values of these coefficients.
5. Provide more explanation for this statement: "Despite these advances, the trade-offs between data-only training and physics-informed training remain poorly characterized". Now it is a little bit vague.
6. Authors say "In conjunction with standard practice, the equations are nondimensionalized to discard the effects of physical units e.g. density and viscosity". Probably, "discard" is wrong word.  In nondimensionalization, the effects of density and viscosity are not discarded. Instead, they are encapsulated within dimensionless numbers like the Reynolds number.
7. See W5. Please explain more about dataset pruning procedure.
8. Figure 3 shows comparison at a "representative timestep of the simulation". But the title of the picture above sounds like "Comparison for u at timestep 0 | MSE: 0.000000...". Authors should provide explanation and fix this issue.
9. Justify choosing SOAP optimizer (why not to choose any other optimizer?).
10. Probably a typo on page 8: "Figure 3 presents both the lowest data loss achieved and the average loss across epochs. " But there are not such values represented on Figure 3.
11. See W7.

---

### Official Review · Reviewer_vYFJ · 2025-10-31

**Soundness:** 2
**Presentation:** 3
**Contribution:** 2
**Rating:** 4
**Confidence:** 4

**Summary:**

The paper completes MeshGraphNets with soft physics constraints to improve surrogate modeling of Navier-Stokes equation. This improvement might be helpful in data-limited settings.

**Strengths:**

*	Solid discussion on nonconvex optimization. The designed optimizer converges fairly fast.
*	The topic of reducing the size of datasets is in fact important.
*	Accuracy near 1% achieved for pressure

**Weaknesses:**

*	The use of physics-informed loss functions in combinations with MeshGraphNets appeared earlier in several works, [1,2], what dims originality of the presented paper.
*	Just one modelling example was presented.
*	Velocity streamlines not illustrated and velocity error non analysed.
*	The title in the PDF (Effects of soft physics constraints…) is different from that in the OpenReview system (From Numerical Solvers to Graph Surrogates…).
*	The acronym “iff.” may not be clear for some readers.
*	“Figure 3 presents both the lowest data loss achieved and the average loss across epochs”. There are no loss functions in Fig. 3.


[1] Würth, T., Freymuth, N., Zimmerling, C., Neumann, G., & Kärger, L. (2024). Physics-informed MeshGraphNets (PI-MGNs): Neural finite element solvers for non-stationary and nonlinear simulations on arbitrary meshes. Computer Methods in Applied Mechanics and Engineering, 429, 117102. https://doi.org/10.1016/j.cma.2024.117102

[2] Zhang, H., Jiang, L., Chu, X., Wen, Y., Li, L., Liu, J., Xiao, Y., & Wang, L. (2025). Combining physics-informed graph neural network and finite difference for solving forward and inverse spatiotemporal PDEs. Computer Physics Communications, 308, 109462. https://doi.org/10.1016/j.cpc.2024.109462

**Questions:**

See weaknesses. Other questions:
* Why pressure not involved in the data loss term (L_data)?
* How exactly L_momentum and L_mass were defined?
* How long the training took in terms of hours? Try to compare versus data modelling with a numerical method.
*	It would be interesting to see evolution of the term||u_actual – u_predicted|| versus time. Same for pressure.
*	Can the trained model be reused to solve the equation on another grid or at other Reynolds? Same question for other boundary conditions and domain/obstacle shapes.
*	How well the conservation law, div V = 0, holds on the learned data?

---

### Official Review · Reviewer_QAWR · 2025-11-01

**Soundness:** 2
**Presentation:** 1
**Contribution:** 1
**Rating:** 2
**Confidence:** 4

**Summary:**

The authors propose to augment MGNs with a physics-informed loss derived from the finite volume method (FVM) residuals of the Navier–Stokes equation + continuity equation, enforcing soft conservation of mass and momentum. The resulting total loss combines data loss with physics residuals, weighted by tunable coefficients. To improve training efficiency under data scarcity, they pair this approach with the SOAP optimizer (Shampoo + Adam preconditioning), along with a theoretical analysis showing that appropriate learning-rate scaling maintains convergence guarantees under static data reduction.
Using OpenFOAM-generated datasets for incompressible 2D cylinder flow (Re = 100), the authors show that:
* adding the soft physics loss reduces number of epochs to converge by up to 33% even when the dataset is reduced by 20%
* model accuracy improves by up to 7.4%

**Strengths:**

* results are provided as practical outcomes
* theoretical evaluation of optimizer with static reduction and schedule scaling
* avoidance of interpolation errors through cell-centered discretization

**Weaknesses:**

* the approach lacks novelty (even practical)
* evaluation is restricted to laminar low-Reynolds 2D cylinder flow which is not the most complicate setting for Navier-Stokes eq.
* the obtained improval is small and made be in the range of the errors of its assessment
* the errors of obtained metrics were not estimated
* no ablations and hyperparameter search
* no computational expenses provided
* no reproducible code

**Questions:**

* How your approach differs from other physics-informed GNNs with FV loss?
* How the time per epoch changed compared to baseline MGN?
* How your approach will deal with turbulent flows? What about space varying viscosity?

---

### Official Review · Reviewer_DU8H · 2025-11-03

**Soundness:** 1
**Presentation:** 2
**Contribution:** 2
**Rating:** 2
**Confidence:** 4

**Summary:**

The authors propose a way to integrate physics loss, derived from Finite Volume Method (FVM) residuals, into a MeshGraphNet (MGN) framework to improve data efficiency and convergence. The motivation appears to originate from the attempt to strike a balance between data-driven vs. physics-driven (PINN-style), which, if achieved, could be a significant step towards advancing the CFD-AI domain.

**Strengths:**

1. I appreciate the authors systematically demonstrating the improvements in convergence speed and accuracy under multiple dataset reduction levels.

2. I also appreciate the theoretical proof that under standard smoothness and bounded-variance assumptions, the SOAP optimizer with the scaled learning-rate schedule achieves the same asymptotic convergence rate as full-data training, under certain conditions.

**Weaknesses:**

1. **Generalizability concerns**: The authors only demonstrate performance across one dataset, raising concerns about whether the performance gain consistently prevails across other datasets and varying geometries. I encourage the authors to consider benchmarks such as airfoils (AirfRANS) and those considered in the MeshGraphnet paper.

2. **Lack of sufficient discussion on the dataset**: The generation of the validation dataset is not clearly discussed. The data generation subsection should discuss various choices, such as why the authors considered unit inlet velocity and Re = 100. In addition, it should also discuss the discretization and important statistics such as the number of nodes, edges, etc., to provide an idea about the scale of the dataset.

3. **Lack of clearly defined problem statement**: Is the author's objective to predict the next-time step or predict a roll-out, as done in MeshGraphnet? If it is to predict only the next time step, that has limited practicality.

4. **Lacking baselines**: Showing improvement over one method (meshgraphnet) is unprecedented in an ML paper. There are many GNN-based methods out there that could benefit from the authors' physics-informed loss and learning rate scheduling, such as FVGCN [Li et al. 2023 in the paper], CFDGCN[2], FV-informed GCN [3], and BSMS-GNN[4]. The authors should consider these baselines to demonstrate that their approach could enhance the existing GNN-based CFD methods in terms of data need and convergence.

5. **Confusing statements**: " For the 80% dataset size, the physics-informed model reached convergence in 33% fewer epochs compared to the baseline, representing the largest observed gain in training efficiency." - What was your stopping criterion that determined this number (33%)?

6. Other minor issues: Figure 2 does not have a professional look. Titles in the subfigures have missing closing brackets.

[1] AirfRANS: High Fidelity Computational Fluid Dynamics Dataset for Approximating Reynolds-Averaged Navier-Stokes Solutions, NeurIPS'22.

[2] Combining Differentiable PDE Solvers and Graph Neural Networks for Fluid Flow Prediction, ICML 2020.

[3] Finite Volume Features, Global Geometry Representations, and Residual Training for Deep Learning‑based CFD Simulation, ICML 2024.

[4] Efficient Learning of Mesh-Based Physical Simulation with Bi-Stride Multi-Scale Graph Neural Network, ICML 2023.

**Questions:**

1. Why is Fig. 3 showing a comparison of only one variable (velocity u-component) at timestep 0? What about other variables, e.g., v and pressure? Could you also add the baseline Meshgraphnet's prediction as well to provide a holistic comparison of their predictive quality?

2. Seek weaknesses 3 and 5.

---

### Meta-Review · Area_Chair_7omg · 2026-01-06

**Summary:**

The reviewers raises several major concerns, the main concerns are the following:

1. Generalizability concerns (by reviewer DU8H, 4MZT)

2. Lacking baselines: Showing improvement over one method (meshgraphnet) is unprecedented in an ML paper (by reviewer DU8H).

3. Lacking novelty (by reviewer QAWR)

4. Lack of clarity (by all reviewers).

After reading the paper, I agree with the reviewers' comments.

**Reviewer Concerns:**

The authors did not address the concerns during the rebuttal.

**Reviewer Scores:**

The scores will remain the same since the authors did not provide rebuttals.

---

### Decision · Program_Chairs · 2026-01-26

Reject